# Mechanism of the small ATP-independent chaperone Spy is substrate specific

Rishav Mitra[1,2], Varun V. Gadkari[3], Ben A. Meinen[1,2], Carlo P. M. van Mierlo [4], Brandon T. Ruotolo [3] &
James C. A. Bardwell[1,2✉]

ATP-independent chaperones are usually considered to be holdases that rapidly bind to non-
native states of substrate proteins and prevent their aggregation. These chaperones are
thought to release their substrate proteins prior to their folding. Spy is an ATP-independent
chaperone that acts as an aggregation inhibiting holdase but does so by allowing its substrate
proteins to fold while they remain continuously chaperone bound, thus acting as a foldase as
well. The attributes that allow such dual chaperoning behavior are unclear. Here, we used the
topologically complex protein apoflavodoxin to show that the outcome of Spy's action is
substrate specific and depends on its relative affinity for different folding states. Tighter
binding of Spy to partially unfolded states of apoflavodoxin limits the possibility of folding
while bound, converting Spy to a holdase chaperone. Our results highlight the central role of
the substrate in determining the mechanism of chaperone action.

[1] Howard Hughes Medical Institute, University of Michigan, Ann Arbor, MI, USA. [2] Department of Molecular, Cellular, and Developmental Biology, University of Michigan, Ann Arbor, MI, USA. [3] Department of Chemistry, University of Michigan, Ann Arbor, MI, USA. [4] Laboratory of Biochemistry, Wageningen University, Wageningen, The Netherlands. ✉email: jbardwel@umich.edu

Topologically complex proteins often populate misfolded intermediates that act as kinetic traps[1]. Such intermediates often expose hydrophobic surfaces that make them prone to aggregation. ATP-dependent molecular chaperones like GroEL-GroES rescue trapped intermediates and facilitate substrate folding[2]. In contrast, ATP-independent chaperones generally bind very tightly to non-native substrates and in doing so prevent protein aggregation but are not thought to directly facilitate substrate refolding[3]. The simple designations of "foldase" or "holdase" may underemphasize the microscopic structural heterogeneity of chaperone–substrate complexes[4]. Substrates bound to ATP-independent chaperones such as trigger factor, SecB, and the sHsps can adopt a wide range of conformations ranging from near-native to unfolded states[5–7]. We have shown that the ATP-independent chaperone Spy not only binds protein folding intermediates of substrate proteins such as Im7 and SH3 but also allows for the folding of substrate proteins while they remain chaperone bound[8,9]. Spy loosely binds to different folding states of these model substrate proteins. Although the folding rate constant for these two simple substrates decreases with increasing concentrations of Spy, it does not go to zero at saturating Spy concentrations where essentially all the substrate molecules are chaperone bound[10], evidence that folding of the substrate occurs while it chaperone-bound. In this mechanism, substrate release is not a prerequisite for substrate folding. We have found that the folding-while-bound mechanism is dependent on relatively weak chaperone–substrate interactions[9]. Variants of Spy that bind SH3 with stronger affinity than does wild-type (WT) Spy significantly slow the folding of bound substrate[9]. The substrates of Spy tested so far have been small, topologically simple, all β-sheet or all α-helical model proteins. We wanted to test if the folding-while-bound model is generalizable to larger substrates with complex topologies such as those found in the vast majority of proteins. Here, we chose apoflavodoxin from *Anabaena* PCC 7119 (AnFld) and *Azotobacter vinelandii* (AzoFld), both well-studied folding models[11,12] that populate kinetically trapped off-pathway intermediates[13,14]. Their α/β topology is both ancient and frequently found in common folds such as TIM barrels, Rossman folds, FAD/NAD(P)-binding domains, and P-loop containing hydrolases. The goal of this study was to test whether the folding-while-bound mechanism adequately describes the effect of Spy on the folding of proteins with complex topologies using the flavodoxin-like fold as an example. Our results show that Spy's mechanism is surprisingly substrate specific. In the case of apoflavodoxin, Spy traps denatured polypeptides in a non-native state, thereby inhibiting flavodoxin folding. This dual functionality of Spy as both a folding-while-bound chaperone and a holdase rests on opposing thermodynamic requirements. The folding-while-bound paradigm requires that Spy bind the various folding states of the substrate weakly. In contrast, a holdase binds non-native states tightly such that folding transitions are not feasibly thermodynamically. We show that the mechanism of action of the chaperone Spy is dictated by the difference in binding affinities for partially unfolded states of different substrates to Spy. This work highlights the need to shift from monolithic enzyme-inspired views of molecular chaperones toward more complex models that incorporate the possibility of multiple substrate-specific mechanisms.

## Results

### Spy inhibits the folding of apoflavodoxin by kinetically trapping it in a non-native state. Spy belongs to an emerging class of chaperones that allow protein folding while the substrate remains

bound to the chaperone[10]. The substrates studied in detail so far, namely Im7 and SH3, are small proteins (10 and 7 kDa, respectively) with very simple 3D structures (just α-helices or just β-strands, respectively). We wondered if the folding-while-bound mechanism of Spy applies to more complex substrates that include both α-helices and β-strands and have a more complex folding pathway than the substrate proteins tested so far. To investigate this, we chose the apoflavodoxin AnFld as our substrate, as its native state has an α–β parallel topology and as apoflavodoxin exhibits a more complex folding pathway than either Im7 or SH3 in that it folds via a three-state triangular mechanism with an essentially off-pathway intermediate (Fig. 1a)[11]. In this mechanism, most of the molecules in the intermediate conformation need to unfold prior to refolding into the native conformation. A small fraction of molecules can directly fold from the intermediate to the native state.

First, we verified that the previously established triangular folding mechanism for flavodoxin folding is functional under our buffer conditions (40 mM HEPES-KOH pH 7.5, 100 mM NaCl), which we considered to be more physiological than the previously used buffer (50 mM MOPS pH 7.0) (Fig. S1a–d)[8,11]. We studied the refolding of AnFld in the presence of Spy by monitoring the tryptophan fluorescence in a stopped-flow fluorimeter upon diluting urea-denatured AnFld into refolding buffer containing increasing concentrations of Spy, as described previously[8]. We observe two kinetic phases of AnFld folding; a major kinetic phase, where the amplitude is proportional to the fraction of denatured molecules that directly fold to the native state (N), and a minor phase whose amplitude is proportional to the fraction of molecules that transiently populate the off-pathway intermediate and therefore need to unfold before productive refolding[11]. In the presence of Spy, the observed rate constants ($k_{obs}$) of both phases decrease substantially until at the highest Spy concentration (~9 μM), a single folding phase remains that has a small $k_{obs}$ value (0.1 s$^{-1}$) (Fig. 1b, c). This indicates that Spy significantly slows the folding rate of AnFld, like for Im7 and SH3. However, both Im7 and SH3 can fold to the native state, albeit slowly, even at saturating Spy concentrations[8,9]. In contrast, binding to Spy prevents folding of AnFld almost completely, as evidenced by the dramatic decrease in the folding amplitude upon increasing Spy concentration (Fig. 1b). At Spy concentrations higher than ~9 μM, the amplitude of the fluorescence change was extremely small, indicating that most of the denatured AnFld molecules are kinetically trapped in a folding-incompetent bound state. Both the unfolded (U) and intermediate (I) states of AnFld are less fluorescent than the native state[11]. Since AnFld in the presence of Spy is substantially less fluorescent than it is in the native state, we reason that Spy kinetically traps AnFld in a non-native state that is partially or completely unfolded (we term this state U*). It appears that Spy binds very rapidly to AnFld and prevents folding since the fluorescence decreases within the ~25 ms dead time of the stopped-flow instrument. The interaction of Spy and the U* state is tight with a dissociation constant ($K_D$) of 0.35 μM (Fig. 1d). The affinity of Spy for AnFld$^{U*}$ in folding experiments is ~13-fold tighter than observed for Spy binding to the intermediate and unfolded states of Im7 ($K_D = 4.7$ μM for both) and ~8-fold tighter than that between Spy and unfolded SH3 ($K_D = 2.9$ μM), both of which fold while bound to Spy[8,9]. The kinetic trapping of AnFld in the Spy-bound U* state due to these high-affinity interactions apparently necessitates release from Spy before productive folding can occur.

### Rapid binding of Spy is coupled to partial unfolding of apoflavodoxin. Having shown that Spy inhibits the folding of AnFld, we wondered if Spy interacts with AnFld's native state. Holdase chaperones are known to preferentially bind to non-native unfolded or misfolded proteins[15]. Folding-while-bound

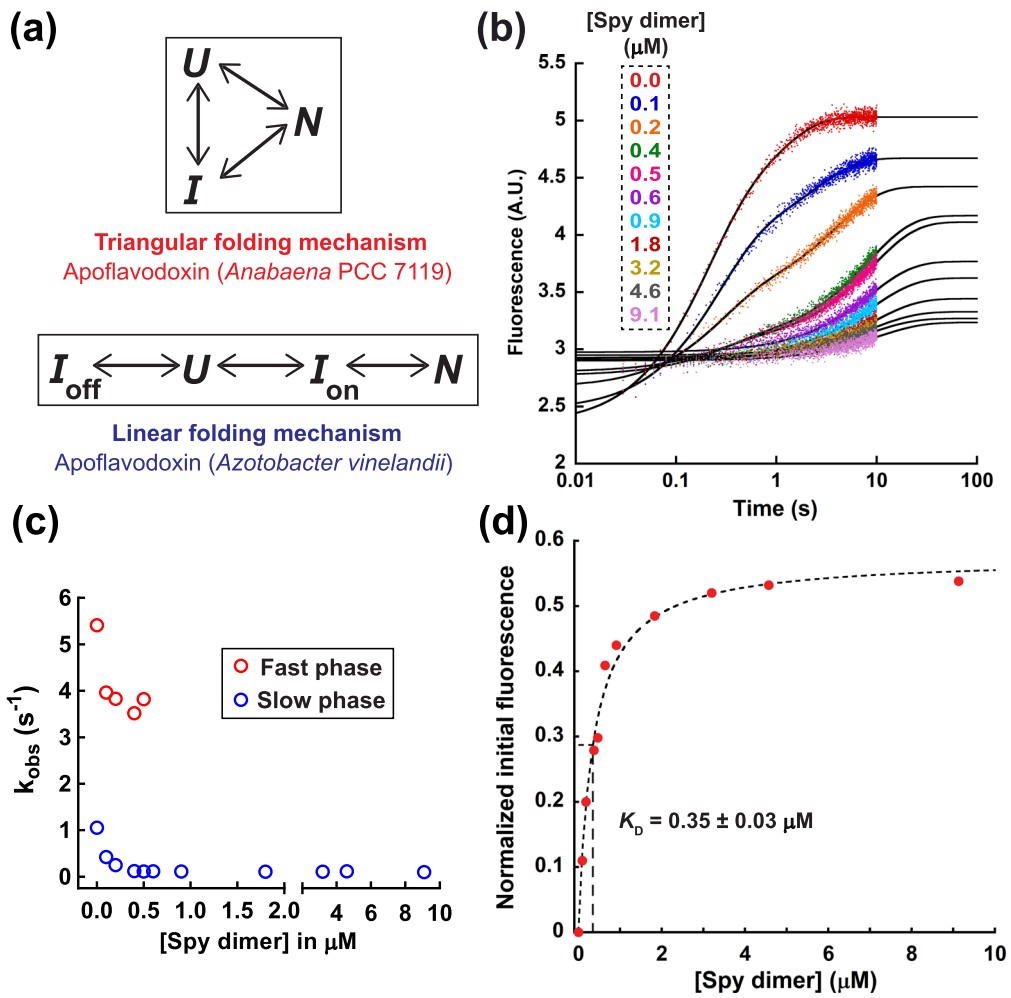

**Fig. 1 Kinetics of apoflavodoxin folding in the presence of Spy. a** Folding pathway of apoflavodoxin from *Anabaena* PCC 7119 (AnFld) and *Azotobacter vinelandii* (AzoFld)[11, 12]. N, U, and I are the native, unfolded, and intermediate states, respectively. In case of AzoFld, $I_{off}$ and $I_{on}$ are the off-pathway and on-pathway intermediates, respectively. **b** Fluorescence traces for the refolding of 0.09 μM AnFld in the presence of various concentrations of Spy dimer (0–9.1 μM after mixing). Time axis is shown in logarithmic scale. AU arbitrary units. The traces of AnFld refolding in the presence of 0–0.5 μM Spy dimer could be fit to a sum of two exponentials, and traces for higher Spy concentrations could be adequately fit to a single exponential function. The absence of the fast phase at higher Spy concentrations most likely indicates that Spy completely blocks the direct folding pathway from unfolded to native AnFld. **c** Plot of the observed rate constants ($k_{obs}$) for AnFld WT folding as a function of Spy dimer concentration. **d** Initial fluorescence values were calculated from the fit of the kinetic datasets. AU arbitrary units. The binding isotherm was plotted with normalized initial fluorescence intensity during AnFld WT refolding in the *y*-axis and fitted to a one-site binding model. All experiments were performed in HN buffer at 25 °C. Value reported is the mean ± s.e.m. of the fit. Source data are provided as a Source Data file.

chaperones, on the other hand, also bind to the native state. Thus, one way to distinguish between holdases and foldases is to determine the relative affinities of the chaperone for native and non-native states. If Spy is functioning as a holdase with AnFld we expect Spy to bind the native state with much weaker affinity than fully and partially unfolded states. We found a binding stoichiometry of 1:1 between AnFld and the Spy dimer using analytical ultracentrifugation (AUC) (Fig. S2a)[8,9]. There is a very rapid decrease in fluorescence upon mixing Spy and AnFld (Fig. 2a). The amplitude of this burst phase reaches saturation at high Spy concentrations, consistent with very rapid binding of Spy to the native AnFld (Fig. S2b). However, the interaction is weak, with an apparent dissociation constant ($K_{D\ app}$) of 40 μM, 114-fold weaker than the $K_D$ for Spy with AnFld$^{U*}$ (Fig. 2b). The $K_{D\ app}$ for native AnFld is similar to that previously reported for the interaction of Spy with native Im7 ($K_D = 20.5$ μM) and SH3 ($K_D = 50$ μM)[8,9]. These comparable

binding affinities for diverse substrates indicates that weak (micromolar) affinity for native substrates is apparently a general property of Spy. This makes sense; tight binding of a chaperone to the native state might well inhibit substrate function. Whether Spy inhibits folding or allows the substrate to fold while bound may therefore depend mainly on the affinity that Spy has for non-native states.

We observed a slow exponential decrease in fluorescence of AnFld in the presence of Spy ($k_{obs} = 0.04$ s$^{-1}$) after the burst phase (Fig. 2b). This decrease is not due to Spy quenching AnFld fluorescence because the final fluorescence decreases hyperbolically with Spy concentration (Fig. S2c). Instead, we suggest that Spy binding leads to a conformational change in AnFld. This conformational change may involve Spy-mediated unfolding of a minor population of native AnFld to a thermal intermediate-like state (I$^T$), which was observed to be less fluorescent than the native state[16]. The native and I$^T$ states have largely similar structures that

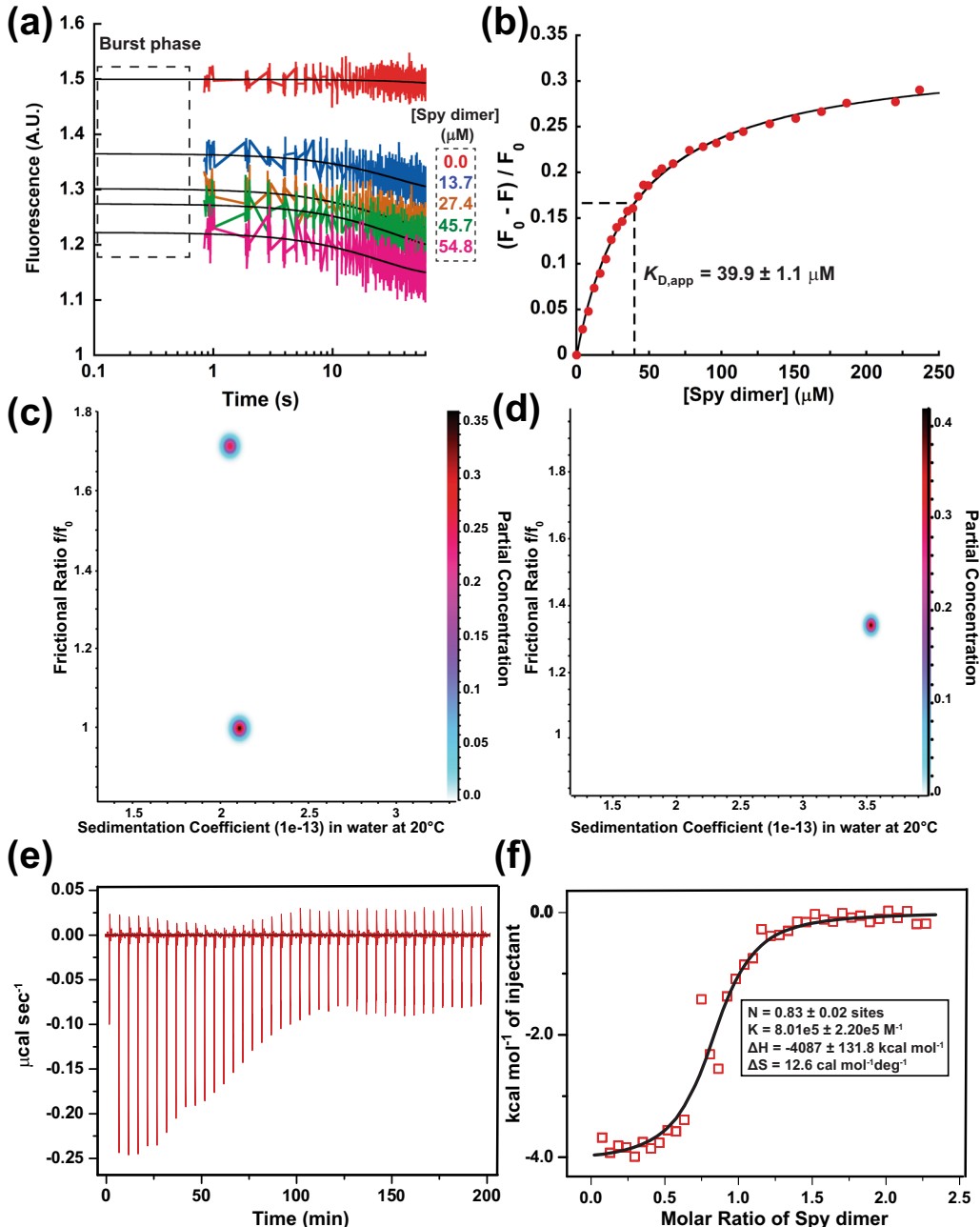

**Fig. 2 Interaction of Spy and native apoflavodoxin. a** Change in intrinsic tryptophan fluorescence of 0.09 μM AnFld WT with time in the absence and presence of various concentrations of Spy (0–54.8 μM after mixing). Time axis is shown in logarithmic scale. AU arbitrary units. The black lines shown for each Spy concentration are best fits of each dataset to a single exponential equation. **b** AnFld WT-Spy binding as monitored by intrinsic tryptophan fluorescence of AnFld. Spy dimer was titrated into 17.2 μM native AnFld WT at 25 °C. The binding isotherm is plotted with change in fluorescence ($F$) as normalized by fluorescence intensity relative to native AnFld fluorescence ($F_0$) in the y-axis. Black line shows the fit to a one-site binding model. **c, d** SV-AUC experiments showing frictional ratios ($f/f_0$) and sedimentation coefficients for **c** AnFld WT alone and **d** AnFld WT in the presence of 2.5-fold excess of Spy dimer in 40 mM HEPES-KOH (pH 7.5), 25 mM NaCl. Data were analyzed by two-dimensional sedimentation analysis (2DSA) followed by analysis with a genetic algorithm, which was further validated by a Monte Carlo analysis. **e, f** ITC analysis for the interaction of Spy and AnFld F98N at 10 °C. The low temperature minimizes enthalpy changes due to the binding-induced unfolding of AnFld. Five hundred and fifty micromolar Spy dimer in the syringe was titrated into 50 μM AnFld F98N in the cell. The thermogram in **e** was integrated and fit to a one-site binding model, as shown in **f** to obtain thermodynamic binding parameters. N stoichiometry of binding, K association constant, ΔH enthalpy change, ΔS entropy change. Values reported are the mean ± s.e.m. of the fit. Source data are provided as a Source Data file.

include two hydrophobic cores formed by α-helices packing onto a central β-sheet. Only the loop connecting β4 and α4 and the long loop splitting the strand β5 are disordered in $I^T$. To further explore the nature of bound AnFld, we used a previously characterized mutant, F98N, that mimics the $I^T$ state[17]. At saturating

concentrations of Spy, the fluorescence of the AnFld WT–Spy complex at 340 nm is 0.33 times that of the fluorescence of AnFld WT alone (Fig. 2b), a value very close to the fluorescence of AnFld F98N (0.39 times that of the WT) (Fig. S2d). This similarity in fluorescence values supports the idea that binding of Spy is

coupled to partial unfolding of AnFld to an $I^T$-like state. Additional support for this proposed partial unfolding comes from studying the interaction of Spy and AnFld in low ionic strength solutions. AUC experiments revealed that in HEPES buffer containing very low salt concentrations (25 mM NaCl), AnFld WT also exists in predominately two conformations that have the same sedimentation coefficient but different frictional ratios ($f/f_0$) (Fig. S2e and Fig. 2c). The major conformation (native state) contributes ~60% of the absorbance signal and has a frictional ratio close to one, typical of globular proteins. The minor conformation contributes ~40% of the signal resembles the partially unfolded $I^T$ in that it is more expanded with a frictional ratio of ~1.7. In the presence of excess amounts of Spy, only a single AnFld WT species is seen (Fig. S2e and Fig. 2d) indicating that the conformational heterogeneity of AnFld seen under non-denaturing conditions is lost upon interaction with Spy. We cannot ascertain whether in the low ionic strength buffer, only the $I^T$ state of AnFld binds Spy and in doing so pulls the equilibrium away from the native state, or whether Spy also binds native AnFld and partially unfolds it to a state that resembles $I^T$. Under physiological salt concentrations however, most of the AnFld molecules populate the native state (Fig. S2f). Isothermal titration calorimetry (ITC) experiments showed that Spy binds the WT and F98N mutant with dissociation constants of 28.2 and 1.8 μM, respectively (Fig. S2g, S2h and Fig. 2e, f). The ~16-fold higher affinity of Spy for the $I^T$ mimic than for WT should shift the equilibrium of Spy-bound AnFld towards the more expanded thermal intermediate. Thus, the molecular basis for binding-induced partial unfolding of native substrates appears to be the higher affinity of Spy to more unfolded conformations[8].

**Spy binds tightly to a partially unfolded mutant of apoflavodoxin.** If the binding of Spy tightly to partially or fully unfolded conformations of AnFld inhibits AnFld refolding, then Spy should have a higher affinity to destabilized mutants of AnFld than it does to the WT protein. We combined two destabilizing mutations in AnFld (L105A and I109A)[18] to generate AnFld 2A. Sedimentation equilibrium experiments showed that this protein is monomeric and soluble (Fig. S3a). Multiple lines of evidence indicate AnFld 2A is partially unfolded. CD spectra indicate the A2 mutant has lost about 30% of its alpha helical content (Fig. 3a and Table S1), though it β-sheet content is unchanged (Table S1). Other evidence that the 2A mutant is partially unfolded include a decrease in intensity and a redshift in tryptophan fluorescence (Fig. 3b) and an increase in ANS (1-anilino-8-napthalene sulfonic acid) binding (Fig. 3c). Sedimentation profiles show that the mutant is expanded relative to the WT protein (Fig. S3a) and urea denaturation profiles indicated that the mutant is highly destabilized (Fig. 3d) and it unfolds more rapidly than WT (Fig. S3b). Limited dispersion in the $^1H$ dimension of the 2D [$^1H$–$^{15}N$] HSQC–TROSY NMR spectra of the mutant between 7.5 and 8.5 ppm indicates that it is mostly disordered, though 48 residues maintain native-like backbone $^{15}N$ shifts (Fig. 3e and Fig. S3c, S3d)[19]. Taken together, our experiments indicate that the AnFld 2A mutant is partially unfolded.

Fluorescence experiments indicate that Spy rapidly binds the 2A mutant with a $K_{D\ app}$ of 0.1 μM (Fig. 3e and S3e). ITC and kinetic unfolding experiments indicate that AnFld 2A exists in two binding-competent conformations that interact with Spy with comparable affinities of 0.38 and 0.23 μM (Fig. S3f and S3b). These affinities measured for AnFld 2A are very similar to the 0.4 μM $K_D$ found for Spy interaction with the U* state of AnFld WT (Fig. 3f). Collectively these observations support our model that Spy kinetically traps AnFld in a partially unfolded state by rapid and tight binding, thus functioning as a holdase.

**Spy forms a compact chaperone–substrate complex with conformationally heterogeneous apoflavodoxin.** To probe the interaction between Spy and AnFld, we employed native mass spectrometry (MS) which is capable of detecting non-covalent protein–protein interactions in the gas phase. Native MS is particularly well-suited for the analysis of mixtures such as co-incubated proteins and their complexes, as each population is separated by mass[20]. When coupled to ion mobility (IM), native IM-MS enables gas phase structural measurements of the mass-resolved proteins and complexes. The IM arrival time distributions (ATD) of individual ions can be converted to rotationally averaged collision cross-section distributions (CCSD). The centroid of these distributions can be reported as the rotationally averaged collision cross section ($^{DT}CCS_{N2}$, CCS). CCS is a charge independent structural parameter that correlates with the surface area of the ion[21]. It is sensitive to domain-level rearrangements that occur in solution prior to ionization. For instance, a rigid protein with a single solution conformation exhibits a Gaussian ATD/CCSD, whereas a dynamic protein exhibits a multimodal or extended ATD/CCSD[22].

Using native IM-MS, we first compared WT AnFld and the destabilized 2A mutant. The CCSs of AnFld 2A are similar to those of AnFld WT, indicating that both sample a similar structural ensemble (Fig. S4b and Table S2). The small difference observed in CCS can be attributed to the small shift in mass of 2A vs. WT. The broad CCSD of both AnFld's (Fig. 4 and Fig. S4b), ranging from 18 to 20 nm$^2$ for the compact states, and 20 to 45 nm$^2$ for the extended unfolded states, indicates that the proteins are structurally heterogeneous in solution. The highest charge states exhibit collision cross-sections equal to those observed for proteins nearly two- to three-fold the size of apoflavodoxin[23,56]. Such conformational heterogeneity is expected as AnFld is destabilized in low ionic strength buffers like the 20 mM ammonium acetate solution used in these experiments[16,24]. The use of this buffer is fortuitous as it gives us an opportunity to detect the native state and otherwise sparsely populated partially folded states of AnFld. The degree of protein folding can also be assessed by inspection of the charge state distribution (CSD) of ions. Natively folded compact proteins with rigid structures have a CSD spanning 2–4 charge states. In contrast, both AnFld proteins analyzed in this work have a broad multimodal CSD spanning 14 charge states, including "completely unfolded" higher charge states (12+ to 19+), some "partially unfolded" charge states (9+ to 11+), and some "folded", compact charge states (6+ to 8+) (Fig. S4) confirming that these proteins populate various conformations in solution. The unfolded 9+ to 19+ charge states account for 15 ± 2% of total AnFld 2A signal intensity, while only accounting for 1.5 ± 0.6% of total AnFld WT signal intensity (Fig. S4a). Although AnFld WT and AnFld 2A both apparently sample the same structural ensemble, the equilibrium of AnFld 2A is clearly more shifted towards the unfolded states compared to AnFld WT.

Next, we analyzed complexes formed between Spy, and both AnFld variants. Three CSDs were observed between 2000 and 4000 $m/z$ corresponding to AnFld, Spy and Spy in complex with AnFld (Fig. 4a, b and Table S2). The Spy:AnFld complexes, containing either AnFld WT or AnFld 2A, consistently exhibit lower CCSs than the larger more unfolded forms of AnFld ($z = $ 15+ to 19+).

Since Spy does not undergo major changes in structure and dynamics upon substrate binding[25,26], we attribute the changes in CCS upon complex formation to changes in the conformation of AnFld upon Spy binding. Measurements of intermolecular paramagnetic relaxation enhancement effects in the Spy–Im7 complex by NMR spectroscopy showed that binding

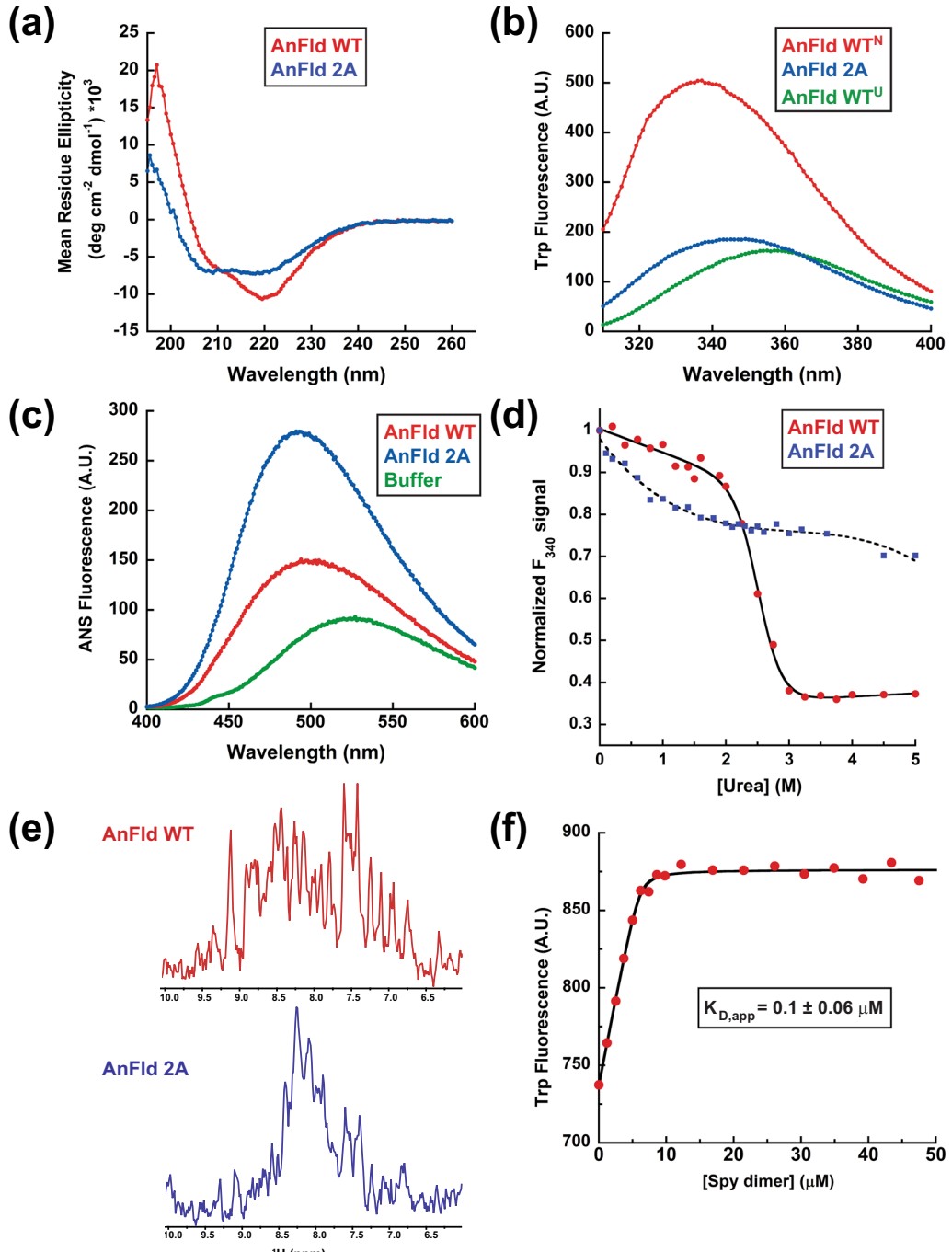

**Fig. 3 Characterization of the AnFld2A mutant and its interaction with Spy. a** Far-UV CD spectra of AnFld WT (red) and AnFld2A mutant (green) in 0.1 M potassium phosphate buffer (pH 7.5) at 25 °C. **b** Intrinsic tryptophan fluorescence emission spectra of the native states of AnFld WT (green), AnFld 2A mutant (blue), and denatured AnFld WT in 5 M urea (red) in HN buffer at 25 °C. **c** Fluorescence emission spectra of ANS alone (red) in HN buffer and in the presence of AnFld WT (green) or AnFld 2A mutant (blue) in HN buffer at 25 °C. The fluorescence intensity of ANS increases 2.8-fold upon binding the mutant compared to the WT. **d** Urea-induced denaturation of AnFld WT (red) and AnFld 2A mutant (blue) monitored by tryptophan fluorescence emission at 340 nm in HN buffer at 25 °C. The data were normalized to the fluorescence signal ($F_{340}$) of each protein in the absence of urea. The denaturation curve of the WT protein was fit to a two-state unfolding model (solid black line) and that of AnFld 2A was fitted to a third-degree polynomial (broken black line). **e** $^1$H dimension of the 2D [$^1$H-$^{15}$N] HSQC–TROSY spectra of 0.2 mM AnFld WT (red) and 2 A (blue). **f** Titration of Spy dimer to 10 µM AnFld 2A as monitored by the intrinsic tryptophan fluorescence. The black line shows the fit of the fluorescence intensity as a function of Spy dimer concentration to a tight binding model. A.U. arbitrary units. Value reported is the mean ± s.e.m. of the fit. Source data are provided as a Source Data file.

of Spy induces compaction of a dynamic ensemble of unfolded Im7 L18A L19A L37A[26]. The compaction of an unfolded client in complex with Spy is proposed to facilitate intramolecular contacts necessary to fold while bound[26,27]. Collectively, our

results from spectroscopy-based kinetic experiments and IM-MS reveal that such client compaction may also stabilize kinetically trapped states that are unable to fold in a chaperone-bound form.

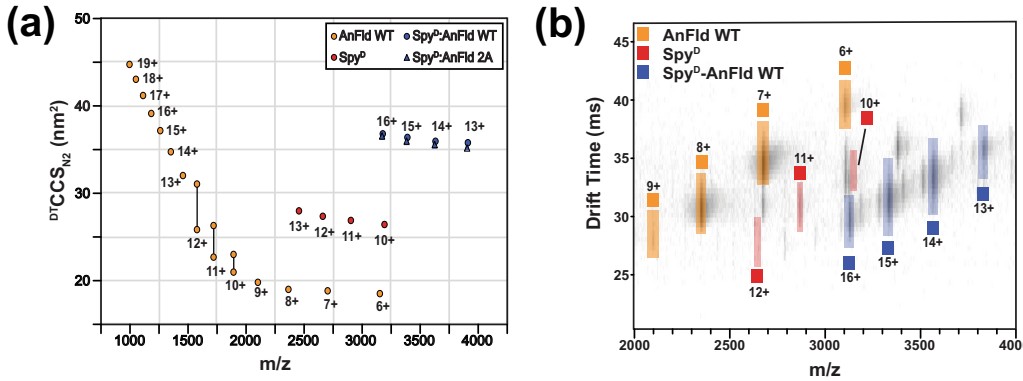

**Fig. 4 Native IM-MS to study AnFld and its complex with Spy. a** Collision cross-sections of AnFld WT (orange circles), Spy (red circles), Spy:AnFld WT (blue circles) versus m/z. The complex of Spy:AnFld 2A (blue triangles) is also plotted for comparison of the complexes. **b** Ion mobility mass spectrum of AnFld WT (orange) co-incubated with SpyD (red). Complexes of SpyD:AnFld WT are highlighted in blue.

**Chaperone-binding surface of apoflavodoxin is composed of dynamic regions implicated in folding.** Using NMR spectroscopy, we set out to characterize the binding surface of AnFld at the residue level. Upon the addition of sub-stoichiometric amounts of unlabeled Spy, we observed a loss of signal intensity, as quantified by the peak heights in the (2D) [$^{15}$N, $^1$H]–TROSY NMR spectra of [$U$-$^2$H, $^{15}$N, $^{13}$C]-labeled AnFld (Fig. S5a). Considering both the increase in size upon complex formation (size of apo-AnFld is ~19 kDa and that of the AnFld-Spy complex is ~50 kDa) and the micromolar affinity of the interaction, we reasoned that line broadening can be caused by increased tumbling time and/or increased chemical exchange in the μs-ms time scale. We attempted to map the binding surface of AnFld for Spy by plotting the ratio of peak intensities of AnFld residues in the absence and presence of 0.75× unlabeled Spy. Residues were considered to be part of the interaction site if they underwent greater than 2/3 of the average loss in peak intensity upon Spy binding (Fig. 5a). These residues largely colocalize on one surface of native AnFld (Fig. 5b, c and Fig. S5b). However, we cannot eliminate the possibility that there are additional binding sites on AnFld as chemical shift assignments were unavailable for stretches of contiguous residues G9–S17 at the phosphate-binding site and C54–Y70 at the FMN-binding site[28].

We note that several residues in the two loops, 90–100 and 120–139, that were found to interact with Spy based on their intensity ratios, have previously been found to be unstructured in the otherwise natively folded thermal intermediate of AnFld (Fig. S5c)[29]. Binding of Spy to these dynamic regions on AnFld may help stabilize the partially disordered AnFld in a high-affinity complex. This is consistent with our native IM-MS experiments, which indicate that the AnFld-Spy complex is more compact than unfolded conformations of unbound AnFld that exist in solution. Interestingly, Monte Carlo simulations using a nucleation-growth model showed that the folding nucleus of AnFld comprises the loop 90–100 and the segment 120–160 that includes α4, β5, most of the C-terminal helix α5 and connecting loops, in other words almost completely overlaps with the binding surface for Spy[30]. While the implications of this finding are unclear, we speculate that Spy may kinetically trap refolding AnFld in the non-native U* state by tightly binding to a folding nucleus that is formed early during folding.

**Spy kinetically traps the misfolded molten globule of another apoflavodoxin.** We have shown that Spy's ability to inhibit the folding of Anabaena flavodoxin apparently results from its tightly binding to partially unfolded states. Given the striking differences between Spy's mechanism of action for AnFld and simple substrates like Im7 and Fyn SH3, we wished to test the holdase activity of Spy with apoflavodoxin from another species, *Azotobacter vinelandii*, that populates a stable molten globule intermediate[12]. Molten globules are a class of compact folding intermediates that lack persistent native tertiary structure but possess a significant amount of native-like secondary structure[31]. The flavodoxin from *Azotobacter vinelandii*, which we term AzoFld, shares 47% sequence identity, and is structurally very similar (RMSD of 0.6 Å) to the flavodoxin from *Anabaena* with which we did all of the proceeding experiments (Fig. S6a). AzoFld folds via a complex four-state mechanism that involves both the formation of a stable off-pathway molten globule intermediate (MG$_{off}$) that acts as a kinetic trap and an obligate high-energy on-pathway intermediate (I$_{on}$) (Fig. 1a)[12]. The major kinetic phase during refolding reports on the rate-limiting step of unfolding of the molten globule intermediate formed rapidly after dilution from denaturant[12].

The sedimentation profiles of AzoFld alone and in the presence of Spy showed that the Spy dimer forms a 1:1 complex with AzoFld (Fig. S6b). By monitoring the refolding of urea-denatured AzoFld in buffer containing increasing concentrations of Spy, we observed that Spy decreased both the folding rate and yield of AzoFld (Fig. 6a). Similar to our observations with AnFld, the decreasing fluorescence amplitude with increasing Spy concentrations indicates that AzoFld cannot undergo efficient folding when bound to Spy. Kinetic refolding of AzoFld occurs in two phases in the absence of Spy. The minor folding phase disappears in the presence of even low amounts of Spy (>4.6 μM). The $k_{obs}$ for the major folding phase decreased hyperbolically with increasing concentrations of Spy and asymptotically reached the very low value of 0.09 s$^{-1}$ at saturating Spy concentrations (Fig. 6b). The simplest explanation for this observation is that Spy binds more tightly to both MG$_{off}$ and U states than it does to native AzoFld, thereby allowing Spy to inhibit the folding of AzoFld. The small non-zero $k_{obs}$ value suggests that unfolding of MG$_{off}$ can still occur while bound to Spy, albeit at a much slower rate. It is noteworthy that the U and MG$_{off}$ states are indistinguishable species in the absence of denaturant due to a gradual second-order-like collapse transition of the unfolded protein to the MG$_{off}$ state under equilibrium conditions[32]. Given that the MG$_{off}$ state is reported to be aggregation-prone in the presence of cytosol-mimicking crowding agents, binding to such misfolded intermediates is additional strong evidence for Spy having a holdase activity for flavodoxin-fold substrates[33,34].

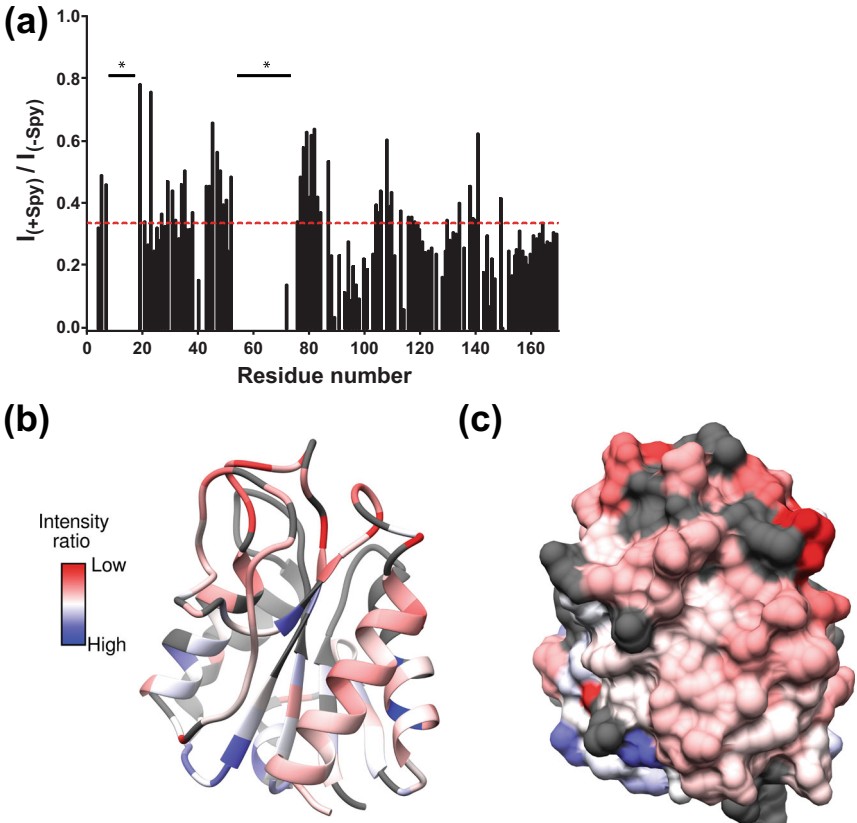

**Fig. 5 NMR spectroscopy to map the Spy binding surface of AnFld. a** A plot of the ratios of intensities of each assigned cross peak in the [$^1$H-$^{15}$N]–TROSY NMR spectra of AnFld WT in the presence of sub-stoichiometric amount of Spy (0.75×) and in the absence of Spy [$I_{(+Spy)}$ and $I_{(−Spy)}$, respectively]. The asterisks denote unassigned residues in the spectra, and the dashed red line shows the cutoff of average peak broadening for all mapped residues. **b**, **c** A red-to-blue color scale has been used to map the NMR peak intensity ratios on the crystal structure of AnFld (Protein Data Bank ID code 1FTG) in **b** ribbon and **c** surface representations. Red and blue represent the highest and lowest intensity ratios in the dataset at 0.784 and −0.002, respectively. A lower intensity ratio, i.e., [$I_{(+Spy)}/I_{(−Spy)}$] is indicative of a higher degree of backbone perturbation in the presence of Spy. The unassigned residues in AnFld are show in gray.

## Discussion

Misfolded and unfolded states of cellular proteins are recognized by ATP-independent holdase chaperones that act by sequestering them to prevent aggregation[3]. Our results with the small bacterial ATP-independent chaperone Spy showed that simple model proteins like Im7 and Fyn SH3 can fold while continuously bound to Spy, provided the different folding states bind the chaperone only weakly[10]. The relative affinity of Spy to the different folding states of the substrate appears to determine whether substrate folding occurs while bound to Spy or not. We previously observed that the Spy mutants Q100L and H96L inhibit SH3 refolding[9]. The mechanistic basis for the unfolding activity of these Spy mutants was an increase in the binding affinity of Spy towards unfolded SH3 ($K_{D, unfolded}$). Furthermore, the affinity of Spy to unfolded SH3 correlates with the folding rate constant in the presence of Spy. These Spy mutants are also potent holdase chaperones that suppress the in vitro aggregation of denatured substrates like aldolase and α-lactalbumin[35]. We postulate that only by binding loosely and with comparable affinity to unfolded, intermediate, and native states, can a chaperone allow folding while bound without interfering with the substrate protein's function. In contrast, binding and sequestration of aggregation-prone denatured states is a hallmark of ATP-independent holdase chaperones[10]. Nanomolar affinities for unfolded polypeptides and folding intermediates have been observed for the interaction of substrates with ATP-independent chaperones such as trigger factor, Skp, SecB, and sHsps[36–39]. The periplasmic holdase

chaperone Skp for instance, binds the unfolded outer membrane protein OmpA with a $K_D$ of 22 nM[37]. The chaperone activity of the human small heat-shock protein Hsp27 is enhanced upon stress-induced phosphorylation[40]. The phosphorylation mimic of Hsp27, S15D/S78D/S82D binds a destabilized T4 lysozyme variant with an apparent affinity of 4 nM[41]. Although the exact mechanism varies, these chaperones generally maintain substrates in an unfolded state, thereby preventing aggregation. That these chaperones bind tightly to non-native states with different degrees of unfolding intuitively supports their holdase activity. In the case of Im7 and Spy, the various Im7 folding states that are bound to Spy can interconvert while chaperone bound apparently at least in part due to the weaker nature of the interaction[25,26]. In the case of AnFld, the interaction of partially unfolded states with Spy is substantially tighter and this tighter binding apparently prevents folding of the chaperone-bound substrate. These observations make intuitive sense; tight binding of the chaperone will hinder conformational transitions in the bound substrate and thus prevent folding while bound. The inability of sHsps to work as folding-while-bound chaperones may be in part due to them relying on short-range hydrophobic interactions to recognize non-native clients[39]. This mode of client recognition also prevents them from binding native proteins, which normally do not expose hydrophobic surfaces. Tight binding to unfolded and intermediate states renders sHsps incapable of spontaneous release and refolding of clients. The chaperone function of sHsps is therefore limited to sequestering aggregation-prone unfolded

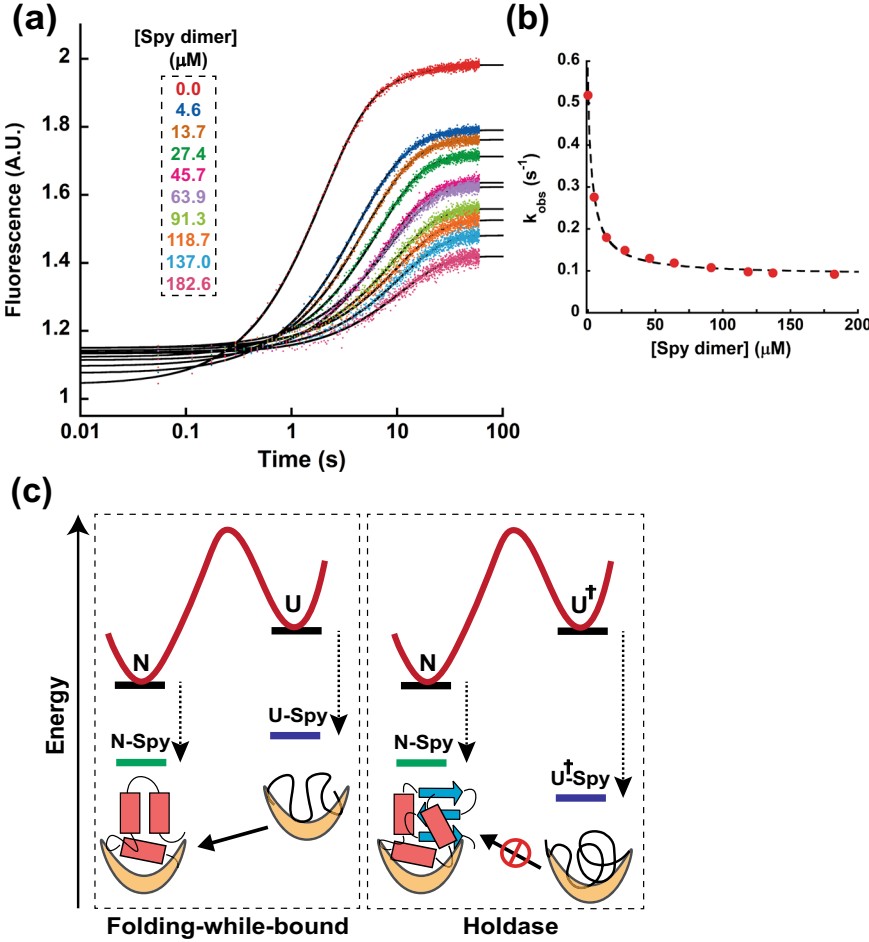

**Fig. 6 Spy inhibits the folding of flavodoxin-fold substrates by acting as a holdase. a** Fluorescence traces of the refolding of 0.1 µM AzoFld C69A in the presence of various concentrations of Spy dimer (0–182.6 µM after mixing) in KP buffer. Time axis is shown in logarithmic scale. **b** A plot of the observed major rate constant for AzoFld refolding ($k_{obs}$) as a function of Spy dimer concentration. **c** Mechanism of substrate-specific action of Spy. Simple substrates can fold while being continuously chaperone-bound because Spy binds the native (N) and unfolded (U) states with weak affinities. Even at saturating concentrations of Spy, folding occurs because the N–Spy complex is the most energetically stable form of the substrate. However, large and topologically complex substrates cannot fold while bound because of Spy's strong affinity for (partially) unfolded states (denoted generically by U†). For such substrates, Spy acts as a holdase by sequestering aggregation-prone U† states. Kinetically, such holdase-like action makes Spy a competitive inhibitor of folding because the U†–Spy complex is the most energetically stable form of the substrate. Source data are provided as a Source Data file.

proteins[38]. Spy on the other hand utilizes mainly long-range electrostatic interactions to recognize and bind unfolded substrates[27]. Although the Spy-unfolded substrate complex is also stabilized by hydrophobic interactions, for substrates like Im7 and SH3, binding is weak and allows the substrate to fold to its native state[27]. Our work with apoflavodoxin shows that the affinity of Spy to partially unfolded substrates is crucial in determining whether Spy's chaperone function mirrors that of sHsps (Fig. 6c).

The substrate-specific mode of action of Spy may enable it to be very effective as an ATP-independent chaperone. Spy has a dual function; it can facilitate protein folding by either allowing folding while bound or it can prevent aggregation by sequestering unfolded substrates or folding intermediates, i.e., by acting as a holdase. Foldase chaperones such as Hsp70, Hsp90, and GroEL exhibit much more complex chaperone cycles that involve cochaperone binding and ATP hydrolysis[3]. Hsp70 chaperones, for instance, depend on J-domain proteins and nucleotide exchange factors for substrate binding and release. These cochaperones modulate Hsp70 function by regulating its ATPase cycle[42]. Unlike these complex and highly evolved foldase chaperones, Spy lacks any known cochaperones or cofactors that can act to modulate

substrate interaction. Instead, Spy's interaction with its substrates is, in some cases (Im7 and SH3), apparently finely tuned by evolution to allow for loose binding and thus folding while bound. In other cases, such as the one studied here with apo-flavodoxin, Spy binds tightly enough to inhibit folding while bound thereby mimicking the action of well-studied holdase chaperones. For topologically complex model substrates like apoflavodoxin, binding tighter to aggregation-prone misfolded states may be the only productive mechanistic possibility in the absence of ATPase activity. Collectively, our results reveal a substrate-specific mechanism for Spy where this chaperone exists with a foot in both the "foldase" and "holdase" worlds and provides interesting insights into both of them.

Since Spy can bind to native proteins with low affinity and non-native proteins with higher affinity, the question arises as to why Spy does not interfere with protein function in the cell and why Spy does not become clogged by high-affinity interactions with folding intermediates. We have previously kinetically and thermodynamically characterized the mechanism whereby Spy binds to, folds and releases it best characterized substrate Im7 (ref. [8]). Periplasmic proteins are highly stable under normal

conditions, which strongly decreases the abundance of periplasmic folding intermediates precluding the need for Spy[43]. However, Spy is overproduced up to 500-fold in response to treatment by protein unfolding agents that lead to the accumulation of (un)folding intermediates in the periplasm. Spy can make up to 25% of the periplasmic protein content[44]. This near stoichiometric abundance of Spy may ensure that enough Spy molecules are available to handle a high client load.

Taken together, this work on Spy and apoflavodoxin and our past work on the effect of Spy on the folding of Im7 and Fyn SH3 establish the central role of the folding landscape of the substrate in determining chaperone mechanisms. Spy can allow simple substrates like Im7 and SH3 to fold while bound because unfolded conformations can transition to the native state on the cradle of Spy. Relatively large and topologically complex substrates like apoflavodoxin, however, will be energetically stabilized in a chaperone-bound unfolded conformation. Our results therefore highlight the importance of substrate-specific mechanisms rather than the paradigmatic roles of ATP-independent chaperones in protein quality control. Although the chaperone field has primarily focused on unified mechanisms that hold true for all substrates, most experimental data for major chaperone families like the Hsp70s and Hsp60s suggest more substrate- and conformation-specific mechanisms[4]. Intuitively, multifunctionality may be advantageous for cellular and organismal fitness as it enables chaperones to bind to different conformational states of substrates.

## Methods

**Protein expression and purification**. The genes for AnFld and AzoFld C69A were codon-optimized for expression in *E. coli* and synthesized by GenScript. The flavodoxin genes were inserted into a pET28b-based vector with an N-terminal His$_6$-SUMO tag by standard restriction digestion and ligation-based cloning using the enzymes *Bam*HI and *Xho*I. The F98N, L105A, and I109A mutations were inserted into AnFld using site-directed mutagenesis with a QuickChange kit (Agilent). The primers used for mutagenesis are listed in Table S3. Plasmid with the spy gene was obtained from a lab collection[8]. The modified pET28b plasmid containing the flavodoxin or spy gene was transformed into *E. coli* BL21 (DE3) cells for protein expression. Cells were grown at 37 °C to early log phase in PEM media containing 100 μg/ml kanamycin and then shifted to 20 °C for induction by addition of 0.1 mM IPTG. After 16 h of protein expression, cells were pelleted by centrifugation for 20 min at 5000 × *g* at 4 °C. The pellet was resuspended in lysis buffer containing 50 mM sodium phosphate, 400 mM NaCl, 10% glycerol, pH 8.0 and 0.05 μg/ml DNase and protease inhibitor cocktail (complete mini EDTA-free; Roche). Following sonication for 8 min on ice, the cell lysate was centrifuged twice at 36,000 × *g* for 30 min at 4 °C to pellet the cell debris. The supernatant was loaded along with 20 mM imidazole onto a Ni-His Trap column that had been equilibrated with water and lysis buffer. The column was washed with lysis buffer containing 30 mM imidazole. His$_6$-tagged protein was eluted by adding lysis buffer containing 500 mM imidazole to the column. Five hundred micrograms of ULP1 protease was added to the elution to cleave the His$_6$-SUMO tag from the protein. The elution was then dialyzed against buffer containing 40 mM Tris, 400 mM NaCl, pH 8.0 at 4 °C overnight. The cleaved His tag was removed following ULP1 digestion and dialysis by passing through a Ni-His Trap column that had been equilibrated with lysis buffer containing 30 mM imidazole. The flow through was collected and 25 mM Tris pH 8.0 buffer was added to dilute the sample fivefold. The sample was then loaded onto a HiTrap Q column and the protein was eluted with a NaCl gradient in 25 mM Tris pH 8.0. In case of flavodoxin, the fractions containing holoprotein were yellow in color and were discarded. The colorless fractions contain purified apoflavodoxin and hence were collected and concentrated. The protein was then passed through a HiLoad Superdex 75 column equilibrated with 40 mM HEPES-KOH (pH 7.5), 100 mM NaCl (HN buffer) for desalting and buffer exchange. The purified protein was then aliquoted, flash frozen in liquid nitrogen, and stored at −80 °C. Expression and purification of AnFld F98N, AnFld 2A, and AzoFld were done with the same protocol. $^{15}$N-labeled AnFld WT and 2A mutant were expressed in M9 minimal media containing $^{15}$NH$_4$Cl as the sole nitrogen source. The protocol of Marley et al.[45] was used to express [U-$^2$H, $^{15}$N, $^{13}$C]-labeled AnFld WT in M9 minimal media prepared in $^2$H$_2$O and supplemented with $^{15}$NH$_4$Cl, $^{13}$C-glucose, and ISOGRO growth supplement (Sigma). The isotopically labeled proteins were purified with the same protocol as that used for unlabeled proteins except that the fractions containing the holoprotein were also collected. Apoflavodoxin was obtained by trichloroacetic acid precipitation to remove the bound FMN[12]. Protein concentrations were determined using the

molar extinction coefficients of 34,100 M$^{-1}$ cm$^{-1}$ for AnFld (WT, F98N, and 2A mutants) and 29,000 M$^{-1}$ cm$^{-1}$ for AzoFld[46,47].

**Analytical ultracentrifugation**. Sedimentation was monitored at 50 krpm by absorbance at 280 nm for AnFld WT and AnFld 2A mutant. For the binding experiments containing Spy and AnFld or AzoFld, sedimentation was performed at 48,000 r.p.m. (180,311 × *g*) and monitored by absorbance at 280 nm. Sedimentation velocity AUC (SV-AUC) was carried out using 450 μl loaded into two-sector epon centerpieces with 1.2 cm path length in an An60Ti rotor in a Beckman Optima Xl-I analytical ultracentrifuge. Measurements were completed in intensity mode. All SV-AUC data were analyzed using UltraScan 4 software, version 4.0 and fitting procedures were completed on XSEDE clusters at the Texas Advanced Computing Center (Lonestar, Stampede) through the UltraScan Science Gateway (https://www.xsede.org/web/guest/gateways-listing)[48]. The partial specific volume (vbar) of the protein samples (Spy, AnFld WT, AnFld 2A mutant, AzoFld) was estimated within UltraScan III based on the protein sequence[49]. Raw intensity data were converted to pseudo-absorbance by using the intensity of the air above the meniscus as a reference, and then edited. Next, two-dimensional sedimentation spectrum analysis (2DSA) was performed to subtract time-invariant noise, and the meniscus was fit using ten points in a 0.05 cm range[50]. The arrays were fit using an S range of 1–8, an *f/f₀* range of 1–4 with 64 grid points for each, 10 uniform grid repetitions, and 400 simulation points. 2DSA was then repeated at the determined meniscus to fit radially invariant and time-invariant noise together using ten iterations. The 2DSA analysis was refined by a genetic algorithm, which helps to define the solutes and to eliminate any false-positive solutions. The distribution between AnFld native state and AnFld intermediate I$^T$ was determined by integrating the peak intensities after performing the genetic algorithm analysis. The results from the genetic algorithm were evaluated using a Monte Carlo algorithm[51].

**Isothermal titration calorimetry**. Thermodynamic analysis of Spy binding to AnFld WT and AnFld F98N was performed by ITC experiments using a MicroCal iTC200 (Malvern Instruments). The proteins were buffer exchanged into assay buffer containing 40 mM HEPES-KOH (pH 7.5), 100 mM NaCl, using a PD10 desalting column. For the WT and F98N mutants, 550 μM Spy dimer was added in the titration syringe and 50 μM AnFld in the cell. For the 2A mutant, 1.2 mM Spy dimer was added in the titration syringe and 100 μM AnFld in the cell. The fitting of thermograms to one-site or two-site models was done using OriginLab that was provided with the instrument.

**Stopped-flow fluorescence experiments**. All stopped-flow experiments were performed in a KinTek SF-300X stopped-flow instrument at 25 °C. The tryptophans of both the flavodoxins were excited at 295 nm and the emitted fluorescence signal was collected using a 320 nm long-pass filter provided with the instrument. Urea dependence of AnFld refolding was monitored by 11.5-fold dilution of 1.04 μM AnFld that was denatured in HN buffer containing 5 M urea into HN buffer containing various urea concentrations in a 1:10.5 mix. AnFld unfolding was monitored in a similar way, except that native AnFld in HN buffer was diluted 11.5-fold in HN buffer containing increasing concentrations of urea between 0 and 5 M. The final concentration of AnFld in all cases was 0.09 μM. Ten to 12 traces of 10 s were acquired for each data point and averaged. For refolding experiments at 0.43–1.9 M final urea concentration, two kinetic phases were observed. However, one exponential was sufficient to fit the traces from experiments at higher urea concentrations. Average traces from unfolding experiments at 2.05–3.38 M final urea concentration were fit to a single kinetic phase, whereas data from experiments at higher final urea concentrations were fit to two phases. AnFld refolding in the presence of Spy was monitored by mixing in a 1:10.5 mix, 1.04 μM AnFld in HN buffer containing 5 M urea with HN buffer containing various Spy concentrations (0–27.4 μM Spy dimer after mixing). Ten to 12 traces of 10 s duration were acquired for each data point and averaged. AzoFld refolding in the presence of Spy was monitored using tryptophan fluorescence intensity as a signal in a similar way as for AnFld, i.e., by 11.5-fold dilution of 1.5 μM AzoFld in 10 mM potassium phosphate pH 6.0 (KP) buffer containing 6 M urea into KP buffer containing various concentrations of Spy (0–182.6 μM Spy dimer after mixing). Each kinetic trace shown is an average of 9–10 traces that were collected for 60 s. For stopped-flow experiments to monitor the interaction of native AnFld and Spy, 1.04 μM AnFld in HN buffer was diluted 11.5-fold into HN buffer containing various concentrations of Spy. Each kinetic trace was an average of 5–6 traces of 50 s duration that were acquired with the auto-shutter switched on to minimize photobleaching during the experiment. The average trace for each data point was fit to a single exponential. In all the experiments with Spy, the background signal of Spy due to its tyrosine residues was removed. This was done by collecting shots of each concentration of Spy diluted into buffer and subtracting the average signal of Spy from the kinetic trace of flavodoxin folding at that Spy concentration. Individual traces were fitted to sums of exponentials in KaleidaGraph (Synergy Software) to obtain observed rate constants.

**Fluorescence spectroscopy experiments**. All fluorescence spectra were acquired on a Cary Eclipse Fluorescence Spectrophotometer. In tryptophan fluorescence-based titrations, binding was monitored by tryptophan fluorescence of

apoflavodoxin; 17.2 μM of AnFldWT or 10 μM AnFld2A in 1 ml of HN buffer was titrated with increasing concentrations of Spy in a 10 mm quartz cuvette containing a magnetic stirrer. After each addition, the contents of the cuvette were manually mixed with a pipette and allowed to equilibrate inside the spectrophotometer for 2–3 min. Spectra were collected at 25 °C with an excitation wavelength of 295 nm and emission wavelengths scanning from 310 to 400 nm. Both excitation and emission slit widths were set to 5 nm. For each addition of Spy, spectra were collected in triplicate over a period of 0.5 min and then averaged. Average spectra for each titration point were corrected for dilution from the titrant additions. For AnFldWT, the fluorescence emission at 340 nm for every titration point was normalized with respect to the emission of flavodoxin alone, plotted as a function of Spy dimer concentration, and the binding isotherm fitted to a square hyperbola equation, Eq. (1). For AnFld2A, the fluorescence emission at 340 nm was plotted as a function of Spy dimer concentration and fitted to a quadratic equation for tight binding, Eq. (2).

$$(F - F_i)/F_i = (B_{max}*[L])/(K_D + [L]) \tag{1}$$

$$F = 0.5 * \left( F_0 \left( C + [L] + K_D - \sqrt{(C + [L] + K_D)^2 - 4 * C * [L]} \right) + I \right) \tag{2}$$

where $F$ is the fluorescence signal, $F_i$ is the fluorescence of AnFldWT in the absence of Spy, $[L]$ is the Spy dimer concentration, $B_{max}$ is the maximum fluorescence change, $K_D$ is the dissociation constant, $F_0$ is a fluorescence correction factor, $C$ is the concentration of AnFld2A, and $I$ is the y-intercept.

The fluorescence emission spectra of native AnFldWT and the AnFld2A mutant were recorded by dissolving the protein in HN buffer, and the spectra of denatured AnFldWT was recorded in HN buffer containing 5 M urea. All the proteins were used at a concentration of 10 μM. The spectra were collected at 25 °C with an excitation wavelength of 295 nm and emission wavelengths from 310 to 400 nm. Both the slit widths were 5 nm. Each spectrum was acquired three times and averaged.

ANS binding assays were performed by monitoring the fluorescence emission spectra of 1-anilino-8-napthalene sulfonate (ANS) in HN buffer in the presence and absence of 5 μM AnFld WT or 5 μM 2A mutant. The final concentration of ANS was 250 μM. ANS was excited with a wavelength of 385 nm and emission was recorded from 400 to 700 nm. The excitation and slit widths were 5 nm. The spectra shown are each an average of five scans.

For chemical denaturation by urea, 10 μM AnFldWT or 5 μM AnFld2A was added to 1 ml HN buffer containing various concentrations of urea ranging from 0 to 5 M and incubated for ~30 min at room temperature. Fluorescence spectra were recorded at 25 °C with excitation wavelength at 280 nm and emission monitored at 340 nm.

**Circular dichroism spectroscopy experiments.** Far-UV CD spectra were acquired on a Jasco J-1500 CD spectrometer. The spectra were recorded in a quartz cell with a path length of 0.1 cm using ~0.2 mg/ml of protein concentration. The spectra were acquired from 260 to 190 nm with a 0.5 nm data interval, 50 nm/min scan speed, and at 25 °C unless otherwise specified. Five scans were collected and averaged. The mean residue ellipticity (MRE) was calculated using Eq. (3)

$$MRE = (MRW * \Theta)/(10 * d * c) \tag{3}$$

where MRW is the mean residue weight, $\Theta$ is the observed ellipticity (degrees), $d$ is the path length (0.1 cm), and $c$ is the protein concentration in mg/ml. The average CD spectra were deconvoluted with the BestSel algorithm to determine the secondary structure of the proteins in terms of the percentages of α-helix, β-sheet, turn, and unordered components[52].

**Nuclear magnetic resonance spectroscopy.** The [1H-15N]–TROSY HSQC NMR spectra of AnFld WT and the 2A mutant were recorded using Watergate solvent suppression at 25 °C on a Bruker 600 MHz spectrometer equipped with the (1H/19F)-X broadband CryoProbe Prodigy. [U-15N]-labeled proteins were exchanged into NMR sample buffer (50 mM potassium phosphate buffer (pH 7.5) containing 0.1 M NaCl) using a PD10 desalting column (GE Healthcare). The samples were 0.2 mM of protein dissolved in a mixture of 90% (v/v) NMR sample buffer (in 1H2O) and 10% (v/v) 2H2O. The spectra were acquired using a total of 2048 complex points in t2 and 512 increments in t1 with 8 scans per increment over a spectral width of 9.6 and 2.1 kHz in the 1H and 15N dimensions, respectively. To study the interaction of AnFld and Spy, [U-2H, 15N, 13C]-labeled AnFld and Spy were exchanged into 50 mM potassium phosphate buffer (pH 7.5) without any salt using a PD10 desalting column (GE Healthcare). 15N-TROSY spectra were collected at 25 °C with 0.2 mM AnFld dissolved in 90% (v/v) buffer (in 1H2O) and 10% (v/v) D2O and then titrations were done by addition of appropriate volume of Spy to the NMR tube. A total of 2048 data points in t2 dimension and 256 increments in t1 with 16 scans per increment were acquired. The spectra were processed using NMRPipe suite, analyzed, and plotted in NMRFAM Sparky[53,54].

**Native ion mobility-mass spectroscopy.** Prior to analysis, Spy and AnFld were exchanged from storage buffer into 20 mM ammonium acetate (pH 7.4) by Micro Bio-spin size exclusion spin columns (Bio-Rad). Spy-AnFld complexes were

formed by mixing Spy dimer and AnFld at 1 μM final concentration in 20 mM ammonium acetate solution (pH 7.4) and co-incubating on ice for ~5 min. Aqueous samples were transferred into the gas phase via nanoelectrospray ionization (nESI) using a capillary voltage of 900–1000 V. The ions were analyzed on a modified Agilent 6560 drift tube ion mobility quadrupole time-of-flight mass spectrometer (DTIM-Q-TOF) (Agilent)[55,56]. The drying gas (N2) temperature was set to 25 °C, and the flow was reduced to 1.5 l/min. The instrument was operated with 99.9999% nitrogen gas, a front funnel pressure of 4.52 torr, a trap funnel pressure of 3.80 torr, a drift tube pressure of 3.947 torr, a quadrupole pressure of $2.48 \times 10^{-5}$ torr, and a TOF flight tube pressure of $1.699 \times 10^{-7}$ torr. The IM separation was carried out under low field conditions (~18.5 V/cm), and the ion arrival time distributions were fit to Gaussian functions using CIUSuite2[57]. The centroids of the fit Guassian functions were converted to rotationally averaged collision cross section ($^{DT}CCS_{N2}$) by a single-field calibration using Agilent Tune Mix ions as previously described[58]. The $^{DT}CCS_{N2}$ measurements reported are the average of three technical replicate measurements, and the error reported is the standard deviation. Raw data were analyzed using Agilent IM-MS Browser 10.0, and mMass[59,60].

**Reporting summary.** Further information on research design is available in the Nature Research Reporting Summary linked to this article.

## Data availability
The raw data generated from analytical ultracentrifugation, isothermal titration calorimetry and NMR spectroscopy experiments have been included as Supplementary Datasets 1–3 and also deposited at The Open Science Framework (OSF) (https://doi.org/10.17605/OSF.IO/ZPM75). The IM-MS data are available from figshare repository with identifier 10.6084/m9.figshare.13530872. Source data are provided with this paper.

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

## Acknowledgements

We thank Sheena Radford, Kevin Wu, and Frederick Stull for helpful discussions and for critical reading the manuscript. We thank Ke Wan in the Bardwell laboratory for protein purification. We are very grateful to Debashish Sahu and Minli Xing at the BioNMR Core of the University of Michigan and Bikash Sahoo and Scott Gorman for their help and guidance with NMR experiments. J.C.A.B. is an investigator at the Howard Hughes Medical Institute, which funded this work. V.V.G. is supported by the Agilent Thought Leader Award presented to B.T.R.

## Author contributions

R.M., V.V.G., B.A.M., B.T.R. and J.C.A.B. designed the study; R.M., V.V.G. and B.A.M. performed the experiments; R.M., V.V.G., B.A.M., C.P.M.v.M. and J.C.A.B. analyzed the data; and R.M., V.V.G., B.A.M. and J.C.A.B. wrote the manuscript.

## Competing interests

The authors declare no competing interests.
