## [Peer Review File · Nature Communications]

REVIEWER COMMENTS

Reviewer #1 (Remarks to the Author):

In this paper Mitra and co-workers study in vitro the consequences for folding of the interaction with the small ATP-independent molecular chaperone Spy and a topologically complex substrate such as apoflavodoxin.

In contrast with the results previously obtained and reported with substrates such as Im7 and fyn-SH3, that could fold while bound to Spy, they found that interaction with Spy prevented apoflavodoxin folding due, they propose, to the strong interaction between chaperone and substrate.

The authors conclude that Spy will act as holdase or as foldase depending on the strength of its interaction with its substrates and it is therefore necessary to also consider the latter when describing the mechanisms by which molecular chaperones play their roles in proteostasis.

Although intuitive and therefore not a surprise the conclusion of the paper is not trivial and is of relevance for the fields of protein folding and proteostasis. The paper may therefore be appropriate for publication in Nature Communications provided that certain concerns, some of them important, are addressed and that its content is made more accessible to the wide readership of the journal (see below).

1 - In Figure 1a the authors present the folding pathway of AnF1d as triangular, where all states are connected. However, in their interpretation of the results presented in Figure 2a, they state that the two exponentials correspond to two phases: fast direct folding and a slow indirect phase populating an off-pathway intermediate that must therefore unfold before folding can occur. If I is directly connected to N why must it populate U on pathway to N ?

2 - In page 14 the authors discuss that native WT AnF1d populates two conformations, the actual native state and a minor partially folded state similar to an intermediate of thermal denaturation called I^T that can be stabilized by mutation of Phe 98 to Asn. Upon interaction with Spy this equilibrium is absent and the authors conclude that "... both conformations of AnF1d can bind Spy". In the following paragraph the authors state that this is due to a population shift from N to I^T upon Spy binding, which is in contradiction and much more appropriate. The former sentence should therefore be rephrased.

3 - The characterization of the properties of mutant 2A of AnF1d is superficial. There is no doubt that it is partially unfolded but the authors should deconvolute the CD spectrum shown in Figure 4a (that suggests a substantial loss of beta secondary structure and, potentially, a gain of alpha secondary structure) to obtain insights into that region of sequence has lost/changed structure. The same applies to the NMR spectrum shown in panel e of the same figure: the reader is told that some signals are not visible in the mutant whereas others have native-like chemical shifts but this information is not made available. A figure is therefore needed showing at least the structure of the native state, the residues mutated, the residues with native-like (HN) chemical shifts and those with resonances not apparent in the spectrum.

4 - At the end of page 19 the authors describe an analysis of the interaction between this mutant and Spy by fluorescence and by ITC. Why is the ITC fit to two different species binding to Spy with quite similar affinities ? The authors then state that these these partially folded populations differ only in their degree of unfolding. Where is the evidence for this ?

5 - Although the experiments are well done, analyzed and interpreted the results are not sufficiently well connected in the narrative of the paper, making the story not as compelling as it could be. For example why is it important to study the interaction of Spy with the native state ? What does it tell us

about the main conclusion of the work ?

6 - It is important that authors make an effort to make the paper more appropriate for a general journal such as Nature Communications. The figures are not very appealing and there is hardly any diagram facilitating a mechanistic understanding of the work. As it is it reads more like a paper for publication in a specialized journal with a strong tradition in protein folding such as Journal of Molecular Biology: if the authors aim at reaching a wider readership they should change the paper accordingly.

Reviewer #2 (Remarks to the Author):

This is a very interesting manuscript that fills a gap in our understanding of chaperone proteins. In particular, it provides much needed thermodynamic clues on the functioning of ATP-independent chaperones. Indeed, the fact that some of them, The chaperone Spy, in particular, had previously been observed to accelerate the refolding of small proteins by simply binding them and favouring, by affinity, the conformations along a specific, productive, pathway. Essentially, in order to work, this process needs the free native state to be at a lower free energy than any of the complexes between Spy and the bound (possibly almost native) intermediate states. But of course this is not necessarily realised for each and every protein. There might be proteins, the authors reasoned, such that some bound states might represent minima more favourable than the free native state. In these cases, one should expect folding to be inhibited by the chaperone. This is precisely what the authors found, namely the presence of stable bound states where an unfolded conformation of the substrate is stabilised by binding. Fully consistently, in the presence of an excess of Spy native substrate (here apoFlavodoxin, AnFld) would convert to a (partially) unfolded conformation, as thermodynamics should dictate if the complex was indeed the state of lowest free energy.

As a theorist, I mostly cannot judge/decide about the quality of the experiments, and their reading. I will let the other reviewers, if experimentalists, to judge there.

This work raises a number of important issues both in itself and in a broader perspective.

1) In the very first few lines of the manuscript, the authors mention the "entropic" stabilisation of non-native proteins. I would like them to explain better why entropic rather than enthalpic. I think I can understand it as implying that partially compact states might be, individually, of higher free energy than the native state, but that their multiplicity makes their ensemble more stable than the native state. If this is what they meant, I believe that adding one more sentence in this direction would make it simpler for the readers to build a mental picture of the energy landscape. If it is not, then it is clear that some further explanation would be useful, at least for readers like me.

2) The very last line at page 14 reports that there are two sub-ensembles for WT AnFld in native conditions, one of them similar to a folding intermediate. Upon addition of a 2.5 fold excess of Spy, they both collapse to a spy-bound complex. The authors conclude thus that the two sub-ensembles can bind to Spy. I see a different possibility, namely that the non-native sub-ensemble binds to Spy and that, by equilibrium, this progressively drains the native sub-ensemble until all protein is bound to Spy. Can they exclude this scenario, and if yes, how? I'd like them to elaborate on this issue and, if necessary, to add a couple of lines to the manuscript to make the scenario clearer.

3) Given that Spy is a periplasmic protein, is it reasonable to speculate that no over-sticky proteins are present in that space, that would otherwise clog Spy? What mechanisms could the cell have to avoid such clogging? What kind of evolutionary pressure would the periplasmic proteins feel not to clog Spy?

4) As proteins get longer, their non-native ensemble becomes more and more heterogeneous. In that

respect, could we think of proteins that are folded by spy, but only in a finite number per Spy dimer, on average, because there is a small chance that Spy will encounter an over-sticky conformation, or that the over-sticky conformation forms during the folding-while-bound process? This possibility ties with question #3.

5) How would the findings of this work connect to other ATP-independent chaperones, such as small Heat Shock Proteins? Why shouldn't they work also as folding catalysts, at least for some substrates?

I would really like these points to be discussed by the authors.

Reviewer #3 (Remarks to the Author):

This is an interesting, well executed study that significantly enhances our understanding of the mechanism of action of Spy, with broader implications for other ATP-independent chaperones. Here, the authors have built on their previous work characterising Spy, interrogating how more complex clients are handled. The finding that the activity of Spy as a holdase or foldase is substrate specific is particularly novel, and has been confirmed here using a number of complementary techniques. The data presented are convincing and the paper is generally well-written. I have not identified any major issues with the manuscript and recommend publication after addressing the minor issues below.

I found the description of Spy "converting" between a foldase and holdase (p5), and being intermediate between these two functions (p10,11), to be confused. I would encourage the authors to emphasise throughout that their data suggests that Spy is multifunctional, with its mode of action selected based on the strength of interactions with clients.

Why the authors did chose a 20 mM ammonium acetate concentration for their MS experiments, given that the ionic strength buffer used in their other experiments is significantly different? The authors state that AnF1d is destabilised in low ionic strength buffers, but did the authors acquire spectra at higher salt concentrations to demonstrate that the unfolding observed is due to solution conditions and not gas phase effects?

Have the authors determined k_{on} and k_{off} for the Spy-apoflavodoxin complexes studied here? It would be interesting to understand if the lifetime of the complexes is also playing a role in modulating Spy's mechanism of action.

How many/which residues in the NMR spectrum of the mutant protein were not observed, ie become disordered? How do these relate to the residues in AnF1d that were shown to bind to Spy, given that the authors posit Spy induces structural destabilisation?

REVIEWER COMMENTS

Reviewer #1 (Remarks to the Author):

In this paper Mitra and co-workers study in vitro the consequences for folding of the interaction with the small ATP-independent molecular chaperone Spy and a topologically complex substrate such as apoflavodoxin.

In contrast with the results previously obtained and reported with substrates such as Im7 and fyn-SH3, that could fold while bound to Spy, they found that interaction with Spy prevented apoflavodoxin folding due, they propose, to the strong interaction between chaperone and substrate.

The authors conclude that Spy will act as holdase or as foldase depending on the strength of its interaction with its substrates and it is therefore necessary to also consider the latter when describing the mechanisms by which molecular chaperones play their roles in proteostasis.

Although intuitive and therefore not a surprise the conclusion of the paper is not trivial and is of relevance for the fields of protein folding and proteostasis. The paper may therefore be appropriate for publication in Nature Communications provided that certain concerns, some of them important, are addressed and that its content is made more accessible to the wide readership of the journal (see below).

Response: We thank the reviewer for these kind comments. We have addressed the detailed comments as listed below. To make the paper more accessible to a general audience, we have extensively revised the wording and the figures, streamlining and clarifying the results section and expanding the discussion by considering the more general implications of this work. We left track changes on so these changes are easily visible. We have also added a brand new commissioned artwork figure that we think nicely summarizes the main take home of the paper in cartoon form.

1 - In Figure 1a the authors present the folding pathway of AnF1d as triangular, where all states are connected. However, in their interpretation of the results presented in Figure 2a, they state that the two exponentials correspond to two phases: fast direct folding and a slow indirect phase populating an off-pathway intermediate that must therefore unfold before folding can occur. If I is directly connected to N why must it populate U on pathway to N?

*In the triangular folding mechanism of AnF1d, as described by Javier Sancho's group (Fernández-Recio, J. et al. **Biochemistry** 40, 15234–15245 (2001)), the observed microscopic rate constants are such that the vast majority of the molecules in the intermediate state have to unfold before folding into the native conformation. Thus although it is technically true that the folding is best modelled as a triangular mechanism it is a reasonable to approximate it as a linear mechanism. Given the similarities in the Chevron plot in our experimental buffer and that used by Sancho's group, we assume that the folding mechanism is the same as they postulated. The urea dependence of the slow refolding phase shows a marked upward curvature at low denaturant concentrations. This is characteristic of an unfolding process where the curvature of the minor folding phase can be best explained by postulating a population of an off-pathway folding intermediate. One other possible explanation is dissociation of oligomers but we excluded this as the source of the curvature in experiments shown in **Suppl. Fig. 1**. We have now suitably modified the figure legend to clarify these points.*

2 - In page 14 the authors discuss that native WT AnF1d populates two conformations, the actual native state and a minor partially folded state similar to an intermediate of thermal denaturation called I^T that can be stabilized by mutation of Phe 98 to Asn. Upon interaction with Spy this equilibrium is absent and the authors conclude that "... both conformations of AnF1d can bind Spy". In the following paragraph the authors state that this is due to a population shift from N to I^T upon Spy binding, which is in contradiction and much more appropriate. The former sentence should therefore be rephrased.

We thank the reviewer for pointing this out. We have now rephrased the phrase "both conformations of AnF1d can bind Spy" to read, "indicating that the conformational heterogeneity of AnF1d seen under non-denaturing conditions is lost upon interaction with Spy". We cannot ascertain whether in the low ionic strength buffer, only the thermal intermediate (I^T) state of AnF1d binds Spy and pulls the equilibrium away from the native state, or whether Spy binds native AnF1d and partially unfolds it to the I^T state while in complex. We certainly agree with the reviewer that we cannot exclude the formal scenario, at least at low ionic strength. However, the native and I^T states have largely similar structures that include the two hydrophobic cores formed by α -helices packing onto the central β -sheet. Only the loop connecting β 4 and α 4 and the long loop splitting the strand β 5 are disordered in the thermal intermediate. Therefore, we think it is unlikely that Spy would bind the intermediate but not the native state.

At physiological salt concentrations, AnFld predominantly exists in the native state (see data below). Therefore, we are confident that Spy binds and partially unfolds a minor population of native AnFld at physiological salt concentrations, similar to what we observed for native Im7. (Stull, F. et al. *Nat. Struct. Mol. Biol.* 23, 53–58 (2016)). We have now added this information in the paper (Suppl. Fig. S2(f)).

Two-dimensional sedimentation analysis (2DSA) plot of analytical ultracentrifugation data for AnFld WT in 40 mM HEPES KOH (pH 7.5), 100 mM NaCl shows a single species that has a frictional ratio around one, typical for globular proteins.

3 - The characterization of the properties of mutant 2A of AnFld is superficial. There is no doubt that it is partially unfolded but the authors should deconvolute the CD spectrum shown in Figure 4a (that suggests a substantial loss of beta secondary structure and, potentially, a gain of alpha secondary structure) to obtain insights into that region of sequence has lost/changed structure. The same applies to the NMR spectrum shown in panel e of the same figure: the reader is told that some signals are not visible in the mutant whereas others have native-like chemical shifts but this information is not made available. A figure is therefore needed showing at least the structure of the native state, the residues mutated, the residues with native-like (HN) chemical shifts and those with resonances not apparent in the spectrum.

We have now added the data from deconvolution of the CD spectra of AnFld WT and the 2A mutant in the main paper and in table S1. We have also reworded the description of the NMR results to make it clear that a comparison of the spectra of WT and the 2A mutant allowed us to assign 48 cross peaks in the mutant unambiguously. These peaks are now clearly identified in Suppl. Fig. 3(c) that shows the 2D NMR spectra of the WT and 2A mutant with resonance assignments. We also included Suppl. fig. 3(d) that shows the crystal structure that also shows the residues mutated along with amino acids whose resonances are not apparent in the mutant and/ or the wild type spectra.

4 - At the end of page 19 the authors describe an analysis of the interaction between this mutant and Spy by fluorescence and by ITC. Why is the ITC fit to two different species binding to Spy with quite similar affinities? The authors then state that these partially folded populations differ only in their degree of unfolding. Where is the evidence for this?

As shown below, we were unable to fit the ITC data to a 1-site model. Additionally, in the kinetic unfolding experiments, we observed two unfolding phases for the A2 mutant, which indicates the presence of two populations at equilibrium, in the absence of aggregation. The finding that the urea-dependence of the rate constants for these two phases are different is evidence that they differ in the amount of solvent-accessible surface upon unfolding. The reviewer however is completely correct that our statement that “these partially folded populations differ only in their degree of unfolding” is an overstatement. We have deleted this claim.

5 - Although the experiments are well done, analyzed and interpreted the results are not sufficiently well connected in the narrative of the paper, making the story not as compelling as it could be. For example why is it important to study the interaction of Spy with the native state? What does it tell us about the main conclusion of the work?

We thank the reviewer for pointing this out. We have now extensively reworded the results and discussion to make the paper more accessible to a general audience. We have also reworked a number of the figures and added a summery mode figure that captures the main point of the paper namely that “our results reveal a substrate-specific mechanism for Spy where this chaperone exists with a foot in both the “foldase” and “holdase” worlds and provides interesting insights into both of them”. We now focus on emphasizing the main finding of this work namely that the mode of action of Spy is substrate-specific, thereby enabling it to be so effective as an ATP-independent chaperone. Spy has a dual function, it can facilitate protein folding by either allowing folding while bound or it can function by preventing aggregation by sequestering unfolded and partially folded states, i.e. by acting as a holdase. For chaperones that act by folding-while-bound, binding to the native state is crucial in allowing the unfolded substrate to fold while being held by the chaperone. To distinguish the holdase and foldase paradigms in the context of chaperones like Spy, it is important to directly test whether the chaperone binds native substrates and to determine the relative affinities for native and non-native states. We explicitly discuss this rational in the middle of page 7 under the heading “Rapid binding of Spy is coupled to partial unfolding of apoflavodoxin”

6 - It is important that authors make an effort to the make the paper more appropriate for a general journal such as Nature Communications. The figures are not very appealing and there is hardly any diagram facilitating a mechanistic understanding of the work. As it is it reads more like a paper for publication in a specialized journal with a strong tradition in protein folding such as Journal of Molecular Biology: if the authors aim at reaching a wider readership they should change the paper accordingly.

We thank the reviewer for these comments. As stated above we have now extensively reworded the manuscript to make it more accessible, we have left track changes on to make it easy to see our changes. We have also added a diagram to describe the mechanistic findings of our work and have changed all the figures to make them more accessible.

Reviewer #2 (Remarks to the Author):

This is a very interesting manuscript that fills a gap in our understanding of chaperone proteins. In particular, it provides

much needed thermodynamic clues on the functioning of ATP-independent chaperones. Indeed, the fact that some of them, The chaperone Spy, in particular, had previously been observed to accelerate the refolding of small proteins by simply binding them and favouring, by affinity, the conformations along a specific, productive, pathway. Essentially, in order to work, this process needs the free native state to be at a lower free energy than any of the complexes between Spy and the bound (possibly almost native) intermediate states. But of course this is not necessarily realised for each and every protein. There might be proteins, the authors reasoned, such that some bound states might represent minima more favourable than the free native state. In these cases, one should expect folding to be inhibited by the chaperone. This is precisely what the authors found, namely the presence of stable bound states where an unfolded conformation of the substrate is stabilised by binding. Fully consistently, in the presence of an excess of Spy native substrate (here apoFlavodoxin, AnFld) would convert to a (partially) unfolded conformation, as thermodynamics should dictate if the complex was indeed the state of lowest free energy.

We thank the reviewer for these kind comments!

As a theorist, I mostly cannot judge/decide about the quality of the experiments, and their reading. I will let the other reviewers, if experimentalists, to judge there.

This work raises a number of important issues both in itself and in a broader perspective.

1) In the very first few lines of the manuscript, the authors mention the "entropic" stabilisation of non-native proteins. I would like them to explain better why entropic rather than enthalpic. I think I can understand it as implying that partially compact states might be, individually, of higher free energy than the native state, but that their multiplicity makes their ensemble more stable than the native state. If this is what they meant, I believe that adding one more sentence in this direction would make it simpler for the readers to build a mental picture of the energy landscape. If it is not, then it is clear that some further explanation would be useful, at least for readers like me.

We thank the reviewer for this comment. Indeed, we meant to imply that entropic stabilization of folding intermediates mean that these partially folded conformations, although they are higher in free energy (i.e., less thermodynamically stable) than the native state, can indeed transiently populate during folding. These folding intermediates exist as kinetic traps in the energy landscape owing to their high degree of conformational entropy. We have now rephrased this sentence in the paper to read "Topologically complex proteins often populate misfolded intermediates that act as kinetic traps in their folding landscape"

2) The very last line at page 14 reports that there are two sub-ensembles for WT AnFld in native conditions, one of them similar to a folding intermediate. Upon addition of a 2.5 fold excess of Spy, they both collapse to a spy-bound complex. The authors conclude thus that the two sub-ensembles can bind to Spy. I see a different possibility, namely that the non-native sub-ensemble binds to Spy and that, by equilibrium, this progressively drains the native sub-ensemble until all protein is bound to Spy. Can they exclude this scenario, and if yes, how? I'd like them to elaborate on this issue and, if necessary, to add a couple of lines to the manuscript to make the scenario clearer.

We thank the reviewer for pointing this out. We have deleted the claim that the two sub-ensembles can bind Spy. We have now rephrased the sentence to say "... indicating that the conformational heterogeneity of AnFld under equilibrium conditions is apparently lost upon interaction with Spy." Our feeling is that the reviewer is probably alluding to the possibility of either an induced-fit (IF) mechanism versus a conformational selection (CS) mechanism for how Spy interacts with AnFld. In most cases, these two possible explanations are difficult to distinguish under the low salt concentrations used in the submission. In low ionic strength buffer, AnFld is destabilized enough to populate the thermal intermediate at room temperature. About 40% of AnFld exists in the intermediate (I^T) state under these conditions. The observed partial unfolding of AnFld upon Spy binding can occur via an IF or a CS mechanism as outlined below.

We cannot be certain whether both native and I^T states bind Spy, albeit with different affinities or whether Spy binds I^T exclusively (CS) and by equilibrium, native AnF1d is drained towards the I^T state. We certainly agree with the reviewer that we cannot exclude this scenario, at least at low ionic strength.

At physiological salt concentrations, however, AnF1d predominantly exists in the N state (see AUC data in response to Comment 2 of Reviewer 1). Two-dimensional sedimentation analysis (2DSA) plot for AnF1d WT in buffer with 100 mM NaCl shows that a single species that has a frictional ratio around one, typical for globular proteins.

Therefore, we are confident that Spy binds and partially unfolds a minor population of native AnF1d in an IF-like mechanism at physiological salt concentrations, similar to what we observed for native Im7 (Stull, F. et al. *Nat. Struct. Mol. Biol.* 23, 53–58 (2016)). We have now added this information in the paper.

3) Given that Spy is a periplasmic protein, is it reasonable to speculate that no over-sticky proteins are present in that space, that would otherwise clog Spy? What mechanisms could the cell have to avoid such clogging? What kind of evolutionary pressure would the periplasmic proteins feel not to clog Spy?

We thank the reviewer for this interesting question. We have addressed these ideas in our discussion as follows:

Since Spy can bind to native proteins with low affinity and non-native proteins with higher affinity the question arises as to why Spy does not interfere with protein function in the cell and why Spy does not become clogged by high affinity interactions with folding intermediates. We have previously kinetically and thermodynamically characterized the mechanism whereby Spy binds to, folds and releases its best characterized substrate Im7. Periplasmic proteins are highly stable under normal conditions, which strongly decreases the abundance of periplasmic folding intermediates precluding the need for Spy. However, Spy is overproduced up to 500-fold in response to treatment by protein unfolding agents that lead to the accumulation of (un)folding intermediates in the periplasm. Spy can make up to 25% of the periplasmic protein content. This near stoichiometric abundance of Spy may ensure that enough Spy molecules are available to handle a high client load.

4) As proteins get longer, their non-native ensemble becomes more and more heterogeneous. In that respect, could we think of proteins that are folded by spy, but only in a finite number per Spy dimer, on average, because there is a small chance that Spy will encounter an over-sticky conformation, or that the over-sticky conformation forms during the folding-while-bound process? This possibility ties with question #3.

This is an interesting possibility. Small proteins like Im7 and SH3 are perhaps such good biophysically amenable folding models because they are very aggregation resistant, indicating that they do not populate sticky intermediates while they fold in isolation. Our previously published evidence indicates that Spy does not alter the folding pathway for least these substrates. Thus, these small proteins are unlikely to form sticky folding intermediates while bound to Spy and as a result stick so tightly to Spy as to inhibit folding while bound. We agree with the reviewer that larger proteins do have a higher probability of populating sticky conformations both while folding in isolation and folding while bound to Spy. If these sticky conformations form while bound to Spy this could make these larger proteins more prone to being held so tightly by Spy that folding becomes thermodynamically unfeasible. Presumably, complex substrates like Apoflavodoxin that populate stable misfolded states that also bind to Spy very tightly, inhibit folding while bound. The reasonable result of tight binding to sticky and partially unfolded states is a holdase action that acts to prevent aggregation. This ties into

Spy's action during stress conditions where even normal stable proteins become likely become sticky. By monovalently interacting with sticky substrate proteins Spy likely prevents them from simultaneously interacting with multiple partners, preventing an initiation of proteome-wide aggregation in the periplasm.

5) How would the findings of this work connect to other ATP-independent chaperones, such as small Heat Shock Proteins? Why shouldn't they work also as folding catalysts, at least for some substrates? I would really like these points to be discussed by the authors.

We also thank the reviewer for this interesting question. We have also incorporated these thoughts into our discussion as follows: "We postulate that only by binding loosely and with comparable affinity to unfolded, intermediate, and native states, can a chaperone allow folding while bound without interfering with the substrate protein's function. In contrast, binding and sequestration of aggregation-prone denatured states is a hallmark of ATP-independent chaperones holdase chaperones¹⁰. Nanomolar affinities for unfolded polypeptides and folding intermediates have been observed for the interaction of substrates to ATP-independent chaperones such as trigger factor, Skp, SecB and sHsps^{38–41}. The periplasmic holdase chaperone Skp for instance, binds the unfolded outer membrane protein OmpA with a KD of 22 nM³⁹. The chaperone activity of the human small heat shock protein Hsp27 is enhanced upon stress-induced phosphorylation⁴². The phosphorylation mimic of Hsp27, S15D/ S78D/ S82D binds a destabilized T4 lysozyme variant with an apparent affinity of 4 nM⁴³. Although the exact mechanism varies, these chaperones generally maintain substrates in an unfolded state, thereby preventing aggregation. That these chaperones bind tightly to non-native states with different degrees of unfolding intuitively supports their holdase activity. In the case of Im7 and Spy the various Im7 folding states that are bound to Spy can interconvert while chaperone bound apparently at least in part due to the weaker nature of the interaction^{28,29}. In the case of AnFld, the interaction of partially unfolded states with Spy is substantially tighter and this tighter binding apparently prevents folding of the chaperone-bound substrate. These observations make intuitive sense; tight binding of the chaperone will hinder conformational transitions in the bound substrate and thus prevent folding while bound. The inability of sHsps to work as folding-while-bound chaperones may be in part due to them relying on short-range hydrophobic interactions to recognize unfolded or misfolded clients⁴¹. This mode of substrate recognition also prevents them from binding native proteins, which normally do not expose hydrophobic surfaces. Tight binding to unfolded and intermediate states renders sHsps incapable of spontaneous release and refolding of substrates. The chaperone function of sHsps is therefore limited to sequestering aggregation-prone unfolded proteins⁴⁰. Spy on the other hand utilizes mainly long-range electrostatic interactions to recognize and bind unfolded substrates³⁰. Although the Spy-unfolded substrate complex is also stabilized by hydrophobic interactions, for substrates like Im7 and SH3, binding is weak and allows the substrate to fold to its native state³⁰.

Reviewer #3 (Remarks to the Author):

This is an interesting, well executed study that significantly enhances our understanding of the mechanism of action of Spy, with broader implications for other ATP-independent chaperones. Here, the authors have built on their previous work characterising Spy, interrogating how more complex clients are handled. The finding that the activity of Spy as a holdase or foldase is substrate specific is particularly novel, and has been confirmed here using a number of complementary techniques. The data presented are convincing and the paper is generally well-written. I have not identified any major issues with the manuscript and recommend publication after addressing the minor issues below.

We thank the reviewer for these very kind comments.

I found the description of Spy "converting" between a foldase and holdase (p5), and being intermediate between these two functions (p10,11), to be confused. I would encourage the authors to emphasize throughout that their data suggests that Spy is multifunctional, with its mode of action selected based on the strength of interactions with clients.

We thank the reviewer for pointing this out. We have now inserted additional material, and rephrased much of the discussion to clarify and emphasize these points (we have left track changes on so the reviewer can easily see these changes. Here is one small example from the discussion: "The substrate-specific mode of action of Spy may enable it to be very effective as an ATP-independent chaperone. Spy has a dual function; it can facilitate protein folding by either allowing folding while bound or it can prevent aggregation by sequestering unfolded substrates or folding intermediates i.e. by acting as a holdase."

Why the authors did chose a 20 mM ammonium acetate concentration for their MS experiments, given that the ionic strength buffer used in their other experiments is significantly different? The authors state that AnFld is destabilised in low ionic strength buffers, but did the authors acquire spectra at higher salt concentrations to demonstrate that the unfolding observed is due to solution conditions and not gas phase effects?

Because of the technical limitations of native MS. In native MS ionization, one is limited to volatilize-able buffers such as ammonium acetate; buffers that are more normal interfere with the measurements. However, even using ammonium acetate if the concentration is too high, it can disrupt non-covalent electrostatic interactions in protein-protein or protein-ligand complexes (Gavriilidou et al. Anal. Chem 87, 20 10378-84 (2015)). Thus, we used 20 mM ammonium acetate to preserve the Spy-Fld interaction. Native MS experiments were not attempted in higher salt concentrations.

Have the authors determined k_{on} and k_{off} for the Spy-apoflavodoxin complexes studied here? It would be interesting to understand if the lifetime of the complexes is also playing a role in modulating Spy's mechanism of action.

Unfortunately, we were unable to determine k_{on} and k_{off} for the interaction of Spy and Apoflavodoxin as binding is extremely rapid. The initial fluorescence upon mixing Spy and AnFld is lower than the fluorescence of AnFld alone, suggesting that binding happens within the dead time of our stopped-flow instrument (about 25 msec).

How many/which residues in the NMR spectrum of the mutant protein were not observed, ie become disordered? How do these relate to the residues in AnFld that were shown to bind to Spy, given that the authors posit Spy induces structural destabilisation?

We thank the reviewer for this question. A comparison of the NMR spectra of WT and the 2A mutant allowed us to assign 48 cross peaks in the mutant unambiguously. These cross peaks and the corresponding residues are now shown in Suppl. Figure 3(d) and (e) respectively, and we state in the results "Limited dispersion in the 1H dimension of the 2D [1H-15N] HSQC- TROSY NMR spectra of the mutant between 7.5 and 8.5 ppm indicates that it is mostly disordered, though 48 residues maintain native-like backbone 15N shifts (Figure S3(d), (e)) (Figure 4(e))22" The remaining residues were unassigned. Many but not necessarily all of these are disordered in the mutant. We cannot list those that are disordered because we have not unambiguously assigned them, as many of them overlap in the congested disordered region of the spectra. Additionally, we observed several cross peaks at regions of the spectra where no native peaks were present. We did not attempt to assign these peaks, which may be non-native but not disordered. However, we can unequivocally state that overall the mutant does tend towards disorder because of this clustering effect. We have now added Suppl. figure 3(d) of the crystal structure of the WT protein showing the 48 native-like residues we could assign based on the WT spectra.

REVIEWERS' COMMENTS

Reviewer #1 (Remarks to the Author):

The authors have greatly improved the manuscript and addressed essentially all the points raised in my first review. I am not sure Figure 7 is appropriate for publication in a scientific journal but this is for the editor to decide.

Reviewer #2 (Remarks to the Author):

I am happy with the authors' replies.

Reviewer #3 (Remarks to the Author):

The author have addressed all of my concerns. The revised version of the manuscript more clearly articulates the authors findings and their importance. I recommend that the revised paper be published. I do not think that the new Figure 7 adds to the manuscript, and the authors may wish to consider removing it before publication.